# CDK4/6 Inhibitors Suppress RB-Null Triple-Negative Breast Cancer by Inhibiting Mutant P53 Expression via RBM38 RNA-Binding Protein

**DOI:** 10.3390/cancers17203339

**Published:** 2025-10-16

**Authors:** Jin Zhang, Kexin Wen, Ken-ichi Nakajima, Yang Shi, Xinbin Chen

**Affiliations:** Comparative Oncology Laboratory, Schools of Veterinary Medicine and Medicine, University of California, Davis, CA 95616, USA; kxwen@ucdavis.edu (K.W.); kennakajima@ucdavis.edu (K.-i.N.); ysyshi@ucdavis.edu (Y.S.)

**Keywords:** CDK4/6 inhibitors, RB, mutant p53, RBM38 RNA binding protein, mRNA translation

## Abstract

**Simple Summary:**

Cyclin-dependent kinase 4/6 (CDK4/6) inhibitors, which are designed for targeting retinoblastoma (RB) protein phosphorylation and thereby block cancer cell growth, have been approved by the FDA as frontline targeted therapies for hormone receptor-positive (HR+), HER2-negative breast cancer. This study explores the potential application of CDK4/6 inhibitors in RB-deficient tumors. Indeed, we found that CDK4/6 inhibitors reduce mutant p53 expression and subsequently suppress tumor cell proliferation in both RB-proficient and RB-deficient triple-negative breast cancer (TNBC) cells. We also found that this effect is mediated by the RNA-binding protein RBM38, which is dephosphorylated at serine 195 by CDK4/6 inhibitors, resulting in the inhibition of mutant p53 mRNA translation. Collectively, our findings suggest that mutant p53 may serve as a predictive biomarker for CDK4/6 inhibitor sensitivity in TNBC and potentially other cancers harboring mutant p53.

**Abstract:**

*Background:* Cyclin-dependent kinase 4/6 (CDK4/6) inhibitors have been developed and clinically used as a frontline targeted therapeutic agent for hormone receptor-positive (HR+), HER2-negative breast cancer. However, the efficacy for CDK4/6 inhibitors varies in different types of cancer and thus there is a need to identify new biomarkers that would help predict efficacy and/or resistance. *Methods:* We examined the effect of CDK4/6 inhibitors in both RB-proficient and -deficient triple-negative breast cancer (TNBC) cells. We also examined whether mutant p53 could be a target and/or prognostic marker for CDK4/6 inhibitors in (TNBC). *Results:* We found that CDK4/6 inhibitors suppress mutant p53 expression in both RB-proficient and RB-deficient TNBC cells. We also found that suppression of mutant p53 is responsible for CDK4/6 inhibitors suppressing TNBC cell survival. Mechanistically, we showed that CDK4/6 inhibitors suppress mutant p53 mRNA translation through the RNA-binding protein RBM38. Previously, we showed that when phosphorylated at serine 195, phosphorylated RBM38 interacts with eIF4G on p53 mRNA and promotes p53 mRNA translation. Indeed, we found that CDK4 phosphorylates RBM38 at serine 195, which subsequently enhances mutant p53 mRNA translation. *Conclusions:* Collectively, our findings suggest that mutant p53 could serve as a potential biomarker for the therapeutic efficacy of CDK4/6 inhibitors.

## 1. Introduction

Cyclin D-dependent kinases (CDK4 and CDK6) are critical regulators of G1-S transition by phosphorylating retinoblastoma protein (RB), which then releases E2F transcription factors for DNA replication [1]. As such, CDK4 and CDK6 are frequently overexpressed or aberrantly activated by overexpression of Cyclin D and subsequently promote tumor cell proliferation in various types of cancer, such as colon, breast, and lung cancers [2]. Thus, targeting Cyclin-dependent kinases 4 and 6 (CDK4/6) has become a significant therapeutic strategy, particularly in breast cancer [3]. Currently, three CDK4/6 inhibitors, Palbociclib, Ribociclib, and abemaciclib, have been approved by the FDA to treat hormone receptor-positive, HER2-negative advanced breast cancer [4,5,6]. These agents inhibit the Cyclin D–CDK4/6–RB pathway, preventing phosphorylation of RB, thereby inducing G1 cell cycle arrest in tumor cells. Thus, RB protein expression is often considered a prerequisite for CDK4/6 inhibitor efficacy. However, clinical studies have indicated that RB status alone is insufficient to predict efficacy, and other mechanisms can contribute to RB-independent therapeutic effects and toxicity mediated by CDK4/6 inhibitors. For example, Cyclin E overexpression or CDK2 activity may confer resistance by bypassing CDK4/6 inhibition [7]. In addition, PI3K pathway mutations may crosstalk with CDK4/6 signaling and thereby influence drug sensitivity [8,9]. As a result, trials have been initiated to explore various combinations of CDK4/6 inhibitors with PI3K inhibitors, immune checkpoint inhibitors, and targeted agents to enhance the efficacy of CDK4/6 inhibitors in breast and non-breast cancers [10,11].

Triple-negative breast cancer (TNBC) accounts for 15–20% of all breast cancer cases and is characterized by the lack of estrogen receptor (ER), progesterone receptor (PR), and HER2 expression [12,13]. Due to the lack of hormone receptors, TNBC patients do not respond to standard endocrine therapies or HER2-targeted therapies. Genomic and clinical analyses of TNBC have revealed frequent alterations, including low RB expression and Cyclin E1 amplification [14]. Thus, the therapeutic benefit of CDK4/6 inhibitors in the TNBC patient population remains unclear as RB is consisdered a key determinant of CDK4/6 inhibitor activity. Interestingly, some results from preclinical studies suggest that CDK4/6 inhibitors show activity in p53 mutant or RB-deficient cells [10,15,16], although the mechanism is not well understood. In addition, TNBC is sensitive to CDK inhibitor treatment when combined with other treatments such as PARP inhibitors [17]. Thus, it is possible that CDK4/6 inhibitors have targets other than RB in TNBC and identification of biomarkers would help to identify which TNBC patients may respond well to CDK4/6 inhibitor therapy [18]. 

The tumor suppressor p53 is known as the “guardian of the genome” due to its critical role in maintaining DNA integrity and preventing tumor development [19,20]. p53 functions as a transcription factor to regulate a plethora of genes involved in specific cell responses, such as apoptosis and cell cycle arrest, and thereby controls the cell’s fate. p53 is found to be frequently mutated in TNBC, reaching ~80% [14]. As a result, mutant p53, which is frequently measured and found to correlate with tumor progression and poor prognosis in TNBC, has been considered as a promising target for TNBC [21,22]. Interestingly, preclinical studies have shown that mutant p53 can cooperate with CDK4/6–Cyclin D1–RB signaling to drive tumor growth. For example, mutant p53 protein can transcriptionally upregulate Cyclin D1, the activating partner of CDK4/6, thereby enhancing CDK4/6 signaling [23,24]. Additionally, CDK4 specifically binds to and phosphorylates mutant p53(R249S) at serine 249 and contributes to mutant p53 gain-of-function [25]. Recently, mutant p53 has been suggested to confer resistance to CDK4/6 inhibitors via suppression of durable senescence or apoptosis [26]. Thus, these data suggest a bidirectional crosstalk between CDK4/6 and mutant p53, which is worth further investigation.

Previously, we found that RNA-binding protein RBM38, when phosphorylated at serine 195, interacts with eIF4G on p53 mRNA and promotes p53 mRNA translation [27]. Interestingly, CDK4 and RBM38 are frequently overexpressed in TNBC, which often harbors mutant p53. This observation prompted us to explore the crosstalk among CDK4, RBM38, and mutant p53 in TNBC. Indeed, we found that CDK4 phosphorylates RBM38 at serine 195, which subsequently enhances mutant p53 mRNA translation. As a result, CDK4/6 inhibitors can suppress mutant p53 expression in both RB-proficient and RB-deficient TNBC cells. Together, our data suggest that mutant p53 can be explored as a biomarker of CDK4/6 inhibitors for TNBC and possibly other tumors harboring mutant p53. 

## 2. Results

### 2.1. Both RB-Proficient and RB-Deficient Breast Cancer Cells Are Susceptible to CDK4/6 Inhibitors

To evaluate the effect of CDK4/6 inhibitors on TNBC cell survival, three well-defined TNBC cell lines, which carry mutant p53, were chosen: MDA-MB-231 (RB-proficient), MDA-MB-468 (RB-deficient), and BT549 (RB-deficient). The ER-positive MCF7 cell line, which is RB-proficient and carries WT p53, was chosen as a positive control since MCF7 cells are known to be susceptible to CDK4/6 inhibitors [28]. Palbociclib and Ribociclib are two well-characterized CDK4/6 inhibitors and known to suppress breast cancer cell survival via the RB pathway [29]. As expected, the colony formation assay showed that MCF7 cell survival was markedly suppressed by treatment with Palbociclib or Ribociclib in a dose-dependent manner (Figure 1A). Similarly, RB-proficient MDA-MB-231 cell survival was also suppressed by treatment with Palbociclib or Ribociclib in a dose-dependent manner (Figure 1B). Interestingly, we found that RB-deficient MDA-MB-468 and BT549 cell survival was also suppressed by treatment with Palbociclib (Figure 1C) or Ribociclib at a relatively higher concentration (Figure 1D). These data were consistent with previous observations that RB-deficient TNBC cells are more resistant to CDK4/6 inhibitors [30,31]. These results were also aligned with the IC50 trends for Palbociclib and Ribociclib in these cell lines, as reported in the Genomics of Drug Sensitivity in Cancer (GDSC) database. Nevertheless, these findings indicate that CDK4/6 inhibitors exert a cell-killing effect on RB-deficient TNBC cells via a target(s) other than RB.

### 2.2. Suppression of Mutant p53 Expression Is Likely Responsible for CDK4/6 Inhibitors Suppressing TNBC Cell Survival

Many studies have shown that p53 is frequently mutated in TNBC and mutant p53 is known to be associated with poor prognosis and resistance to CDK4/6 inhibitors in TNBC [21,22]. However, the link between mutant p53 and CDK4/6 inhibitors is not well understood. In this regard, we sought to determine whether CDK4/6 inhibitors can modulate mutant p53 expression in RB-proficient and RB-deficient TNBC cells. Specifically, RB-proficient MDA-MB-231 cells were treated with various amounts of Palbociclib and Ribociclib, followed by measurement of the phosphorylation of RB at Threonine 356 (p-RB) as well as mutant p53. Threonine 356 in RB is a crucial phosphorylation site targeted by CDK4/6, which leads to inactivation of RB growth suppression activity and cell cycle progression [32]. We found that the level of threonine 356 phosphorylation in RB was markedly inhibited by Palbociclib and Ribociclib in MDA-MB-231 cells (Figure 2A,B). Interestingly, we also found that the level of mutant p53 protein was decreased by Palbociclib and Ribociclib in MDA-MB-231 cells in a dose- and time-dependent manner (Figure 2A,B and Appendix A). By contrast, the level of wild-type p53 in RB-proficient MCF7 cells was only moderately decreased by Palbociclib and Ribociclib (Appendix A), consistent with a previous report [33]. Similarly, we found that mutant p53 expression was decreased by Palbociclib and Ribociclib in a dose-dependent manner in RB-deficient MDA-MB-468 and BT549 cells (Figure 2C–F). 

It is well-established that mutant p53-carrying tumor cells, including TNBC cells, are addicted to mutant p53 for their survival [34,35]. To determine whether inhibition of mutant p53 by CDK4/6 inhibitors contributed to growth suppression, mutant p53 was knocked down in MDA-MB-231 and MBA-MB-468 cells, followed by a cell viability assay. As shown in Figure 2G,I, the mutant p53 protein was efficiently reduced by two different siRNAs. Moreover, we found that, consistent with a previous study [36], knockdown of mutant p53 led to decreased cell viability (Figure 2H,J, left mock-treated panels). Interestingly, while Palbociclib was potent in suppressing cell viability of control cells transfected with scrambled siRNA, the cell viability of cells transfected with p53 siRNAs was not further decreased in MDA-MB-231 (Figure 2H) and MDA-MB-468 (Figure 2J). Together, these findings suggest that mutant p53 is a target of CDK4/6 inhibitors and contributes to growth suppression mediated by CDK4/6 inhibitors in both RB-proficient MDA-MB-231 and RB-deficient MDA-MB-468 cells.

### 2.3. Mutant p53 Expression Is Regulated by CDK4 via mRNA Translation

Previous studies indicate that CDK4 and p53 are intricately linked in cellular processes, and their crosstalk has significant implications for cancer development and treatment [26,37]. Additionally, CDK4 has been found to modulate p53 transcriptional activity and moderately decrease wild-type p53 expression [33]. Thus, to demonstrate that the effect of CDK4/6 inhibitors on mutant p53 expression is due to CDK4 inhibition, we sought to examine whether CDK4 modulates mutant p53 expression. Indeed, we showed that the level of mutant p53 protein was increased by overexpression of CDK4 in MDA-MB-231, MDA-MB-468, and BT549 cells (Figure 3A–C). By contrast, upon knockdown of CDK4 by two separate siRNAs, mutant p53 expression was decreased in all three TNBC cells (Figure 3D–F). 

Since both WT and mutant p53 expression are regulated by CDK4/6 inhibitors in this study (Figure 2 and Appendix A), we speculated that CDK4 modulates mutant p53 mRNA translation. To test this, the level of newly synthesized mutant p53 protein, was measured by a biotin-based Click-iT assay in MDA-MB-231 cells transiently transfected with a scramble or CDK4 siRNA for 3 days [38,39]. In this assay, cells are first labeled with the methionine analog L-homopropargylglycine (HPG), which is incorporated into newly synthesized polypeptides, followed by a "click" reaction that conjugates a biotin azide to the alkyne group on HPG, allowing for the detection of biotin-labeled, newly synthesized proteins. Interestingly, we found that the level of newly synthesized p53 protein was markedly decreased by knockdown of CDK4 in MDA-MB-231 cells (Figure 3G). Consistent with this, we found that the level of newly synthesized p53 protein was decreased by Palbociclib in both MDA-MB-468 and BT549 cells (Figure 3H,I). Together, these data indicate that CDK4 is required for mutant p53 mRNA translation.

### 2.4. CDK4/6 Inhibitors Suppress Mutant p53 Expression and Cell Survival via RBM38 RNA-Binding Protein

Previous studies from our group showed that p53 mRNA translation is modulated by the RNA-binding protein RBM38 and eIF4F translation complex [40]. Specifically, phosphorylation of RBM38 at serine 195 enhances its interaction with eIF4G on p53 mRNA and, therefore, p53 mRNA translation [27]. Interestingly, the UACLAN database indicated that RBM38 and CDK4 are frequently overexpressed in TNBCs (Appendix A) [41,42]. We also found that although CDK4 is highly expressed in both wild-type and mutant p53 breast cancers, RBM38 is only highly expressed in breast cancers harboring mutant p53 (Appendix A). These observations let us speculate that CDK4 may phosphorylate RBM38 to enhance mutant p53 expression via mRNA translation in TNBC. To test this, MCF7 cells expressing ectopic RBM38 were treated with or without Plabociclib or Ribociclib, followed by measurement of RBM38, RB, and actin. We would like to note that RBM38 protein is expressed as two polypeptides; the slowly migrating polypeptides are due to phosphorylation at serine 195, called p-RBM38, whereas the quickly migrating polypeptides are under-phosphorylated RBM38, called RBM38 [27]. Interestingly, we found that upon treatment with Palbociclib, the level of p-RBM38 protein decreased (Figure 4A,B), which was accompanied by a slightly increased expression of under-phosphorylated RBM38 protein (Figure 4A,B). Next, we determined whether endogenous p-RBM38 can be inhibited by the CDK4/6 inhibitor by treating MDA-MB-231 cells with various amounts of Palbociclib. We found that the level of p-RBM38 was decreased by Palbociclib in a dose-dependent manner in MDA-MB-231 cells (Figure 4C, RBM38 panel), which was accompanied by decreased expression of mutant p53 (Figure 4C, p53 panel). Consistent with this, we found that ectopic expression of CDK4 in BT549 cells led to increased expression of endogenous p-RBM38. Since p-RBM38 is known to enhance p53 expression [27], we thus sought to determine whether RBM38 plays a role in CDK4/6 inhibitor-mediated repression of mutant p53 expression. To this end, isogenic control and RBM38-KO Mia-PaCa2 cells were mock-treated or treated with Palbociclib, followed by detection of RBM38, mutant p53, and actin. We found that Palbociclib decreased the level of p-RBM38 and mutant p53 in the isogenic control MIA-PaCa2 (Figure 4E, left panel) but had no effect on mutant p53 expression in RBM38-KO MIA-PaCa2 cells (Figure 4E, right panel). Consistently, we found that knockdown of CDK4 was able to inhibit mutant p53 expression in the isogenic control but not in RBM38-KO MIA-PaCa2 cells (Figure 4F). To determine whether RBM38 is required for CDK4/6 inhibitors to suppress cell survival, a colony formation assay was performed and showed that Palbociclib was potent enough to suppress cell survival in isogenic control cells but had little if any effect on cell survival in RBM38-KO cells (Figure 4G).

### 2.5. CDK4 Interacts with RBM38 and Mediates Phosphorylation of RBM38 at Serine 195

Protein kinase is known to interact with its substrate transiently [43]. Thus, reciprocal IP-WB analysis was performed. Briefly, 293T cells were transiently transfected with plasmids expressing Flag-tagged CDK4 and HA-tagged RBM38, followed by IP–Western blot analysis. By using an HA antibody, we immunoprecipitated RBM38 and detected co-precipitated CDK4 by Western blot (Figure 5A), indicating a physical interaction between exogenous RBM38 and CDK4. To verify this, HCT116 cells were transiently transfected with HA-tagged RBM38 and then subjected to IP–Western blot analysis. We showed that immunoprecipitation of HA-tagged RBM38 pulled down endogenous CDK4 (Figure 5B), confirming an interaction between the two proteins. Similarly, HA-tagged RBM38 was detected in anti-CDK4 immunoprecipitates (Figure 5C). 

To verify the interaction between CDK4 and RBM38, an in situ proximity ligation assay (PLA) was performed by using MCF7 cells expressing Flag-tagged CDK4 and HA-tagged RBM38. We found that PLA signals were detected in cells stained with anti-RBM38 to detect RBM38 and anti-Flag to detect CDK4, thus indicating the close association between CDK4 and RBM38 (Figure 5D). No PLA signal was detected in the negative control samples using the anti-Flag or anti-RBM38 antibody alone (Figure 5D). Similarly, PLA foci were detectable when anti-Flag was used to detect CDK4 and anti-HA was used to detect RBM38 for immunostaining (Appendix A). 

To further confirm that CDK4 phosphorylates RBM38, we performed the phos-tag gel assay, which detects protein phosphorylation by inducing a mobility shift in phosphorylated proteins relative to their nonphosphorylated counterparts during electrophoresis [44]. Briefly, 293T cells were transiently transfected with a plasmid expressing wild-type RBM38 or mutant RBM38(S195A), with or without CDK4. We found that upon co-expression with CDK4, the WT RBM38 protein showed a marked increase in the level of p-RBM38 protein, as detected by Phos-tag gel (Figure 5E). By contrast, mutant RBM38(S195A), in which serine 195 is substituted with nonphosphorylatable alanine, did not exhibit detectable phosphorylation and thus was unaffected by CDK4 (Figure 5E). Similarly, we found that in 293T cells transfected with wild-type RBM38, the level of p-RBM38 was increased by CDK4, as measured by standard Western blot analysis (Figure 5F). By contrast, in 293T cells transfected with RM38(S195A), no p-RBM38(S195A) protein was detected regardless of CDK4 expression (Figure 5F). Together, these data indicate that CDK4 interacts with and phosphorylates RBM38 at serine 195.

## 3. Discussion

CDK4 inhibitors have been approved by the FDA for the treatment of hormone receptor-positive (HR+), HER2-negative breast cancer [45]. Although RB protein expression is considered a prerequisite for CDK4/6 inhibitor efficacy, RB status alone is insufficient to predict treatment response. For example, CDK4/6 inhibitors exhibited cytotoxicity in RB-deficient tumors where the canonical CDK4/6-RB pathway is disrupted, suggesting RB-independent mechanisms. Thus, there is a growing need to identify additional biomarkers for CDK4/6 inhibitors to improve patient outcomes or predict resistance. In this study, we found that CDK4/6 inhibitors suppress cell survival in both RB-proficient and RB-deficient TNBC (Figure 1). We also found that mutant p53 expression is suppressed by CDK4/6 inhibitors in a RBM38-dependent manner and that decreased mutant p53 expression is responsible for growth suppression mediated by CDK4/6 inhibitors (Figure 2). Moreover, we found that CDK4 interacts with and phosphorylates RBM38 at serine 195 to enhance mutant p53 mRNA translation (Figure 4 and Figure 5). Together, our findings revealed novel crosstalk among CDK4, RBM38, and mutant p53. Considering that mutant p53 expression is routinely measured by pathologists for diagnosis and that mutant p53 is known to antagonize RB function, we postulate that mutant p53 can be used as a biomarker to evaluate CDK4/6 inhibitor response.

Previously, we found that mutant p53 mRNA translation is modulated by GSK-3β, which phosphorylates RBM38 at serine 195 [27]. Here, we showed that CDK4 modulates mutant p53 mRNA translation (Figure 3) through phosphorylation of RBM38 at serine 195 (Figure 4 and Figure 5). Notably, RBM38 is frequently found to be overexpressed in breast cancer, including TNBC, and associated with mutant p53 status (Appendix A) [46]. Similarly, in hepatocellular carcinoma, CDK4 and RBM38 were also found to be highly expressed in mutant p53-containing cancer (Appendix A). These observations suggest that CDK4 could serve as a kinase for RBM38 and consequently enhances mutant p53 expression. Given the frequent occurrence of mutant p53 in various tumor types and its association with poor prognosis and therapy resistance, targeting this CDK4-RBM38-mutant p53 pathway presents a potential therapeutic strategy. As such, CDK4/6 inhibitors, which are already approved by the FDA, may be further investigated as a treatment option for malignancies harboring mutant p53, either alone or in combination with other targeted therapies.

In summary, we identified mutant p53 as a potential biomarker for the efficacy of CDK4/6 inhibitors and revealed novel crosstalk among CDK4, RBM38, and mutant p53. Thus, further studies are warranted to determine whether patients with TNBC or other cancers with p53 mutation would benefit from taking CDK4/6 inhibitors as an adjuvant agent. 

## 4. Materials and Methods

### 4.1. Reagents

Anti-RB (Cat# sc-74562), anti-pRB (Cat# sc-377527), anti-p53 (Cat# sc-126), and anti-actin (sc-8432) were purchased from Santa Cruz Biotechnology (Dallas, TX, USA). Anti-HA (Cat# 901514) was purchased from Biolegend (San Diego, CA, USA). Anti-Flag (Cat# 80801-2-RR) was purchased from Proteintech (Rosemont, IL, USA). Anti-RBM38 was customized and affinity-purified previously [47]. Anti-CDK4 (Cat# 12790) and anti-GAPDH (Cat# 2118S) were purchased from Cell Signaling Technology (Danvers, MA, USA). Streptavidin magnetic beads (Cat# HY-K0208), Palbociclib (Cat# HY-50767), and Ribociclib (Cat# HY-15777) were purchased from Medchem Express (Monmouth Junction, NJ, USA). RNAiMAX was purchased from Life Technologies (Carlsbad, CA, USA) (Cat#: 13778). The WesternBright Sirius HRP substrate (Cat# K12043-D20) was purchased from Advansta (San Jose, CA, USA). JetPRIME transfection reagent was purchased from Polyplus (Illkirch, France). 

### 4.2. Cell Culture

MCF7, HCT116, Mia-PaCa2, MDA-MB231, MDA-MB-468, BT549, and 293T cells were purchased from the American Type Culture Collection (ATCC). Isogenic control and RBM38-KO Mia-PaCa2 cells were previously generated [48]. HCT116 and MCF7 cell lines expressing HA-tagged RBM38 were generated previously [40,47]. Cells and their derivatives were cultured in Dulbecco’s Modified Eagle Medium (DMEM; Life Technologies) supplemented with 10% fetal bovine serum (FBS; Life Technologies). All the cells were used below passage 25 or within 2 months. As all ATCC cell lines undergo comprehensive testing and authentication, no further authentication was performed for this study. 

### 4.3. Plasmids and Transient Transfection

pcDNA3 vector expressing Flag-tagged CDK4 was purchased from Genescript (Piscataway, NJ, USA). pCDNA3 vector expressing HA-tagged RBM38 was generated previously [47]. To transfect the plasmids to cells, JetPRIME transfection reagent was used according to the user manual.

### 4.4. SiRNA Transfection

SiRNA transfection was performed using RNAiMAX reagent according to the user manual. Briefly, to prepare a lipid and siRNA complex, siRNA at a final concentration of 20 nM was diluted and mixed with RNAiMAX at a 1:1 ratio. The mixture was incubated at room temperature for 5 min and then added to the cells. The sequence for scrambled siRNA was 5′-GCA GUG UCU CCA CGU ACU A-3′. The sequences for CDK4 siRNA were 5′-GGU GAC AAG UGG UGG AAC AUU-3′ and 5′-GAA CUG ACC GGG AGA UCA AUU-3′. The sequences for p53 siRNA were 5′-GCA CAG AGG AAG AGA AUC UUU-3′ and 5′-GAA AUU UGC GUG UGG AGU AUU-3′.

### 4.5. Proximity Ligation Assay (PLA)

PLA was performed by using DuoLink PLA assay kit purchased from Sigma(St. Louis, MO, USA) according to the manufacturer’s protocol. Cells were transfected with indicated plasmid DNAs, fixed with 3.7% formaldehyde, and permeabilized with 0.2% Triton X-100. The fixed cells were blocked and then incubated with primary antibodies at 4 °C for approximately 20 h. The cells were washed and incubated with PLUS and MINUS probes for 1 h at 37 °C. The cells were then incubated with DNA ligase from Promega (Madison, WI, USA) for 30 min, followed by DNA polymerase with a labeled probe for 100 min at 37 °C. The cells were mounted with ProLong Gold mounting medium with DAPI and observed with a Leica SP8 confocal microscope (Wetzlar, Germany) with 40× oil immersion objective. The cell images were processed with FIJI software (Version 1.54p). 

### 4.6. Western Blotting and Immunoprecipitation

Western blotting was performed as previously described (the uncropped blots are shown in Appendix A) [49]. Briefly, cell lysates were loaded on 8–13% SDS–polyacrylamide gels to separate the proteins based on their size. After gel electrophoresis, the proteins were transferred from the polyacrylamide gel to a nitrocellulose membrane. The membranes were incubated with a primary antibody at 4 °C overnight, followed by 4 h of incubation of a secondary antibody. Next, Western Bright Sirius HRP substrate (Advansta, San Jose, CA, USA) was incubated with the membrane, followed by detection using the UVP ChemStudio imager with VisionWorks LS software (Version 8.20.17096.9551) (Analytik Jena, Jena, Germany).

For the immunoprecipitation assay, cells were lysed using IP lysis buffer (50 mM Tris HCl, pH 7.4, 150 mM NaCl, 1 mM EDTA, and 1% NP-40) supplemented with the proteinase inhibitor cocktail (100 μg/mL). The cell lysates were then incubated with 1 μg of control IgG or the antibody of interest along with magnetic protein A/G beads at 4 °C overnight. The next day, the immunocomplex was washed 8 times using IP lysis buffer and subjected to Western blot analysis. 

### 4.7. Click-It Assay Kit

The assay was performed according to the manufacturer’s manual. Briefly, cells were first starved with methionine-free DMEM supplemented with 1% FBS, followed by labeling with L-Homopropargylglycine (HPG), an amino acid analog of methionine, for 30 min, followed by a click reaction to conjugate biotin azide to HPG-incorporated proteins. Cell lysates were then immunoprecipitated with magnetic streptavidin beads, followed by Western blot with anti-p53.

### 4.8. Colony Formation Assay

A total of 2 × 10^3^ cells in triplicates were seeded into each well of a six-well plate. Forty-eight hours after seeding, cells were either mock-treated or treated with various concentrations of Palbociclib or Ribociclib for 48 h, and then the drugs were withdrawn. The cells were then cultured for an additional one or two weeks to allow for colony formation. At the end of the experiment, cells were fixed with a solution of methanol and glacial acetic acid solution (7:1), followed by staining with crystal violet. After staining, the excess crystal violet was removed by washing with tap water, and the plates were air-dried.

### 4.9. CellTiter-Glo Viability Assay

The CellTiter-Glo assay (Promega, Madison, WI, USA) was performed according to the manufacturer’s guide. Briefly, 2 × 10^3^ cells in 100 μL medium were seeded into each well of a 96-well plate in triplicate, followed by mock treatment or Palbociclib treatment for 24 h. At the end of the experiment, 100 μL of CellTiter-Glo Reagent was added to each well and the plate was shaken on an orbital shaker for 2 min to induce cell lysis and then incubated at room temperature for 10 min. The luminescence from each sample was recorded using a SpectraMAX Gemini microplate reader (Molecular Device) (San Jose, CA, USA). The Relative Cell Survival was calculated as a percentage of the Average Luminescence of Treated Cells/Average Luminescence of Control Cells.

### 4.10. Statistical Analysis

Student’s *t* test was used for statistical analysis. *p* < 0.05 was considered significant.

## 5. Conclusions

In conclusion, we found that CDK4/6 inhibitors reduce mutant p53 expression, and subsequently suppress tumor cell proliferation in both RB-proficient and RB-deficient triple-negative breast cancer (TNBC) cells. We also found that this effect is mediated by the RNA-binding protein RBM38, which is dephosphorylated at serine 195 by CDK4/6 inhibitors, resulting in the inhibition of mutant p53 mRNA translation. Collectively, our findings suggest that mutant p53 may serve as a predictive biomarker for CDK4/6 inhibitor sensitivity in TNBC and potentially other cancers harboring mutant p53.

## Figures and Tables

**Figure 1 cancers-17-03339-f001:**
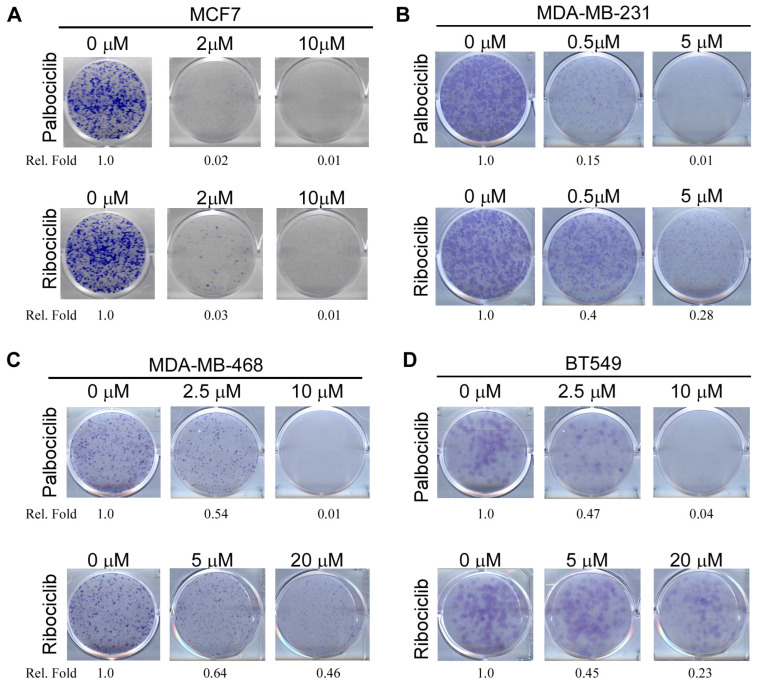
Both RB-proficient and RB-deficient breast cancer cells are susceptible to CDK4/6 inhibitors. (**A**) Colony formation assay was performed with MCF7 cells treated with or without Palbociclib (2 and 10 μM) or Ribociclib (2 and 10 μM) for 48 h, after which the drugs were withdrawn to allow for colony formation. The relative fold change in cell survival is shown below each treatment. (**B**) The experiment was performed the same as in (**A**) except that MDA-MB-231 cells were mock-treated or treated with Palbociclib (0.5 and 5 μM) or Ribociclib (0.5 and 5 μM) for 48 h. The relative fold change in cell survival is shown below each treatment. (**C**,**D**) Colony formation assay was performed with MDA-MB-468 (**C**) and BT549 (**D**) cells mock-treated or treated with Palbociclib (2.5 and 10 μM) or Ribociclib (5 and 20 μM) for 48 h. The relative fold change in cell survival is shown below each treatment.

**Figure 2 cancers-17-03339-f002:**
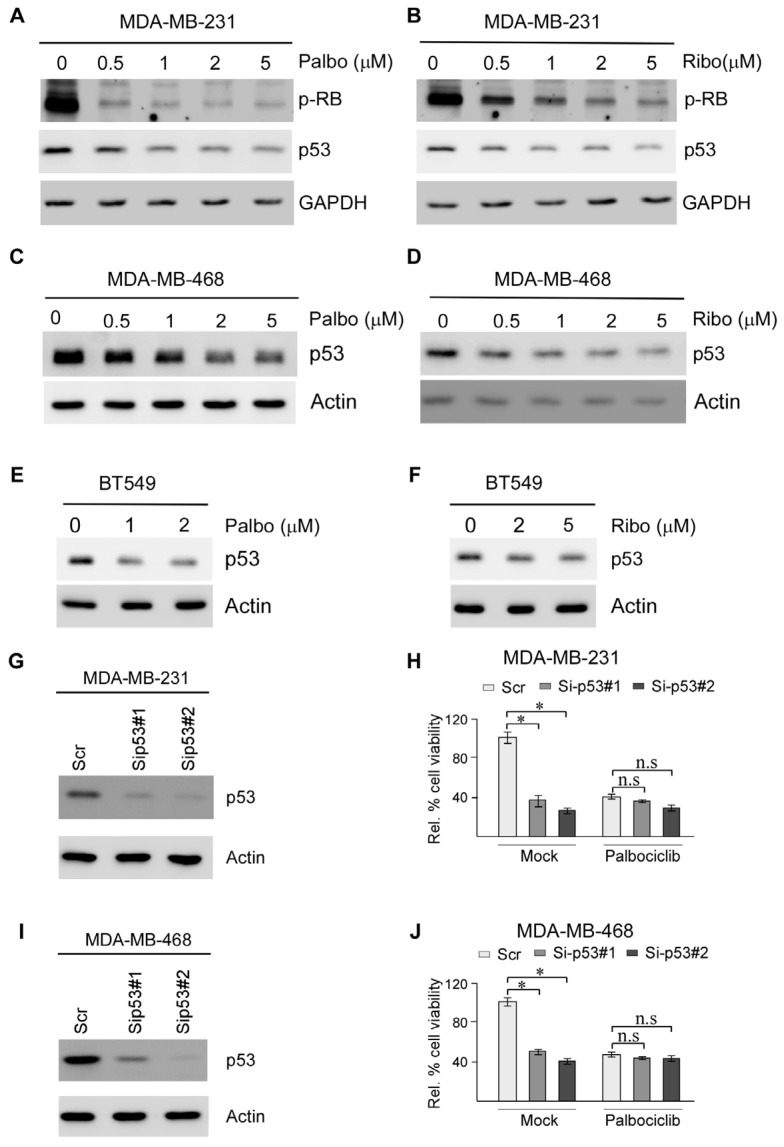
Suppression of mutant p53 expression is likely responsible for CDK4/6 inhibitors sup-pressing TNBC cell survival. (**A**,**B**) MDA-MB-231 cells were mock-treated or treated with Palbociclib (0–5 μM) (**A**) or Ribociclib (0–5 μM) (**B**) for 24 h, followed by Western blot to measure p-RB, p53 and GAPDH. (**C**,**D**) MDA-MB-468 cells were mock-treated or treated with Palbociclib (0–5 μM) (**C**) or Ribociclib (0–5 μM) (**D**) for 24 h, followed by Western blot to measure p53 and actin. (**E**,**F**) BT549 cells were mock-treated or treated with Palbociclib (0–2 μM) (**E**) or Ribociclib (0–5 μM) (**F**) for 24 h, followed by western blot analysis to measure p53 and actin. (**G**,**I**) MDA-MB-231 (**G**) and MDA-MB-468 (**I**) cells were transiently transfected with a scrambled siRNA or p53 siRNAs for 3 days, followed by Western blot to measure the expression of p53 and actin. (**H**,**J**) MDA-MB-231 (**H**) and MDA-MB-468 (**J**) cells were transiently transfected with a scrambled siRNA or p53 siRNAs for 3 days, followed by treatment with or without 5 μM of Palbociclib for 24 h. The cell viability was measured by the CellTiter-Glo luminescent cell viability assay. * indicates *p* < 0.05 by Student’s *t* test; n.s. indicates not significant.

**Figure 3 cancers-17-03339-f003:**
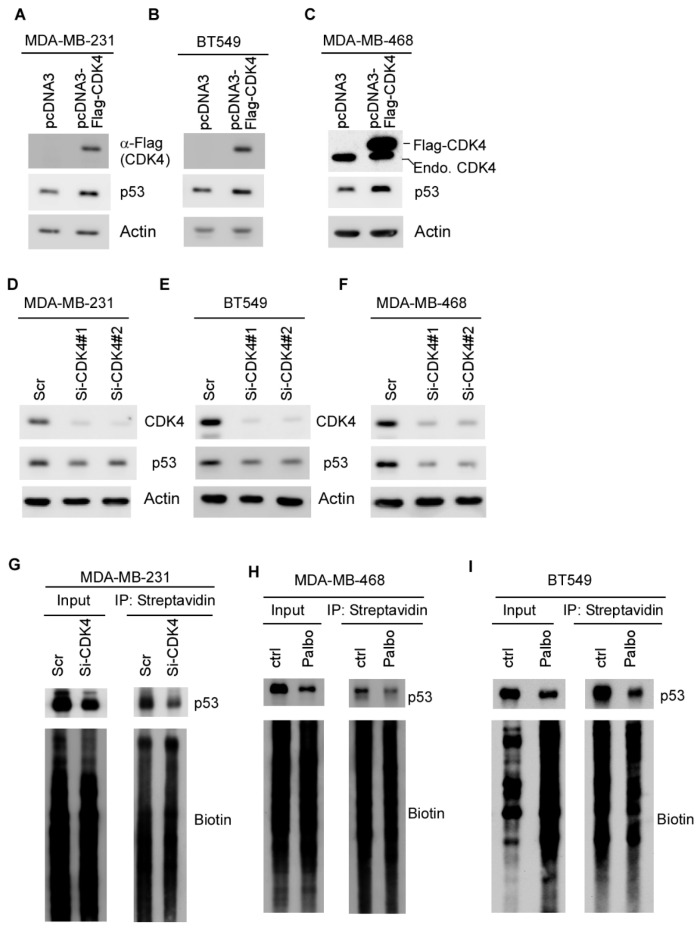
Mutant p53 expression is regulated by CDK4 via mRNA translation. (**A**–**C**) MDA-MB-231 (**A**), BT459 (**B**), and MDA-MB-468 (**C**) cells were transfected with an empty pcDNA3 vector or a vector expressing Flag-tagged CDK4 for 24 h. Cell lysates were then collected and subjected to Western blot analysis using antibodies against CDK4, p53, and actin. (**D**–**F**) MDA-MB-231 (**D**), BT459 (**E**), and MDA-MB-468 (**F**) cells were transiently transfected with scrambled siRNA or siRNAs against CDK4 for 3 days. The cell lysates were then collected to detect CDK4, p53, and actin proteins by Western blot analysis. (**G**) MDA-MB-231 cells transfected with scramble or CDK4 siRNA for 3 days. The level of newly synthesized p53 proteins was measured by performing Click-it assays as described in “Section 4”. (**H**,**I**) MDA-MB-468 and BT549 cells were mock-treated or treated with 5 μM of Palbociclib for 24 h. Click-it assays were then performed to measure the level of newly synthesized p53 proteins.

**Figure 4 cancers-17-03339-f004:**
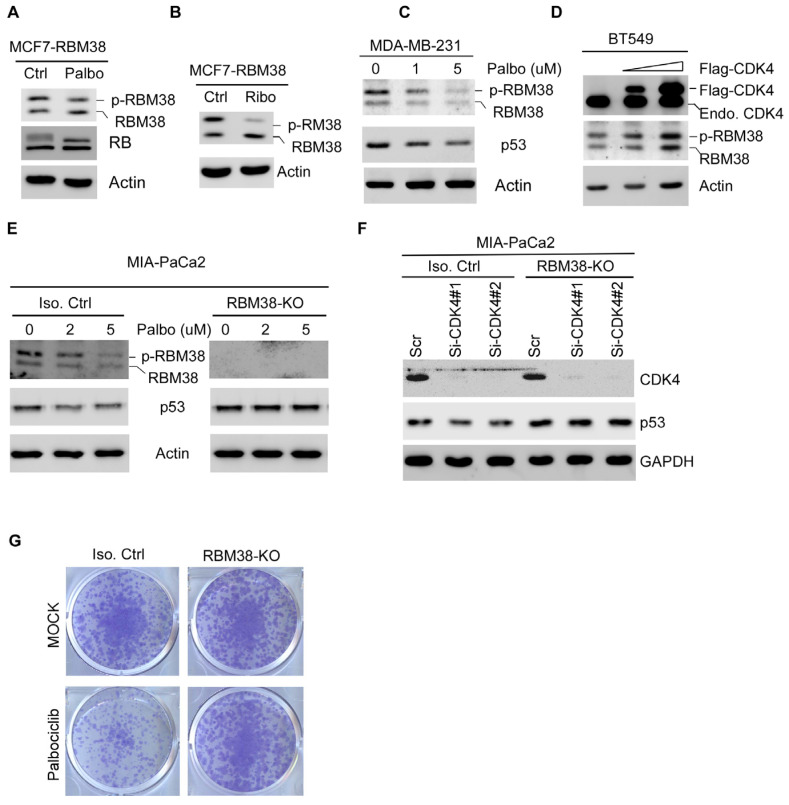
CDK4/6 inhibitors suppress mutant p53 expression and cell survival via RBM38 RNA-binding protein. (**A**) RBM38-expressing MCF7 cells were treated with or without 5 mM of Palbociclib for 12 h, followed by Western blot analysis to measure the level of RBM38, RB, and actin. (**B**) RBM38-expressing MCF7 cells were treated with or without 5 μM of Ribociclib for 12 h, followed by Western blot analysis to measure the level of RBM38 and actin. (**C**) MDA-MB-231 cells were mock-treated or treated with Palbociclib (0–5 μM) for 18 h, followed by Western blot analysis to measure p-RBM38, RBM38, p53, and actin. (**D**) BT549 cells were transiently transfected with a control vector or various amounts of Flag-tagged CDK4-exprssing vector, followed by Western blot using antibodies against CDK4, RBM38, and actin. (**E**) Isogenic control and RBM38-KO MIA-PaCa2 cells were mock-treated or treated with Palbociclib for 24 h, followed by Western blot analysis using antibodies against RBM38, p53, and actin. (**F**) Isogenic control and RBM38-KO MIA-PaCa2 cells were transfected with scrambled siRNA or siRNAs against CDK4 for 3 days, followed by Western blot analysis to measure the expression of CDK4, p53, and GAPDH. (**G**) A colony formation assay was performed. Isogenic control and RBM38-KO MIA-PaCa2 cells were mock-treated or treated with Palbociclib (5 μM) for 48 h, after which the drug was withdrawn to allow individual colony growth for 2 weeks.

**Figure 5 cancers-17-03339-f005:**
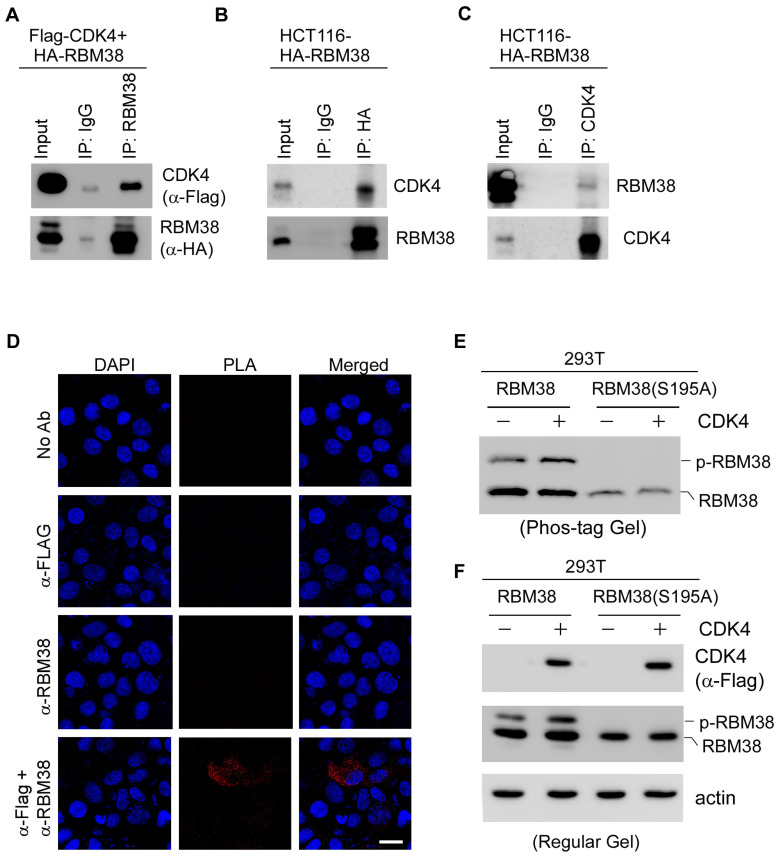
CDK4 interacts with RBM38 and mediates phosphorylation of RBM38 at serine 195. (**A**) 293T cells were co-transfected with vectors expressing Flag-tagged CDK4 and HA-tagged RBM38 for 24 h. Cell lysates were collected and immunoprecipitated with 1 mg of isotype control IgG or anti-RBM38 antibody. The immunocomplex was then subjected to Western blot analysis using anti-Flag to detect CDK4 or anti-HA to detect RBM38. (**B**,**C**) RBM38-expressing HCT116 cells were immunoprecipitated with isotype control IgG or anti-HA (**B**) or anti-CDK4 (**C**). Western blot analysis was performed to detect CDK4 or RBM38. (**D**) MCF7 cells were transiently transfected with Flag-tagged CDK4 and HA-tagged RBM38 for 24 h. The interaction between RBM38 and CDK4 was then detected by PLA, as described in “Section 4”. Scale bar: 20 μM. (**E**) 293T cells were transiently transfected with a vector expressing WT RBM38 or RBM38(S195A), with or without Flag-tagged CDK4, for 24 h. Cells lysates were subjected to a phos-tag gel assay, followed by Western blot with an antibody against RBM38. (**F**) The cell lysates in (**E**) were ran in a regular SDS-Page gel, followed by Western blot with antibodies against CDK4, RBM38, and actin.

## Data Availability

The authors confirm that the data supporting the findings of this study are available within the article, the source data, and its Appendix A.

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
