# Peer review of "CDK4/6 Inhibitors Suppress RB-Null Triple-Negative Breast Cancer by Inhibiting Mutant P53 Expression via RBM38 RNA-Binding Protein"

_cancers, 2025, doi:10.3390/cancers17203339_

Round 1

Reviewer 1 Report

Comments and Suggestions for Authors

In the manuscript entitled “CDK4/6 inhibitors suppress RB-null triple negative breast cancer by inhibiting mutant p53 expression via RBM38 RNA-binding protein”, Zhang et al has found that CDK4 inhibitors suppress mutant p53 expression in TNBC, especially in Rb-deficient TNBC.  Mechanistically, the authors found that CDK4 phosphorylates RNA-binding protein RBM38, which subsequently modulates mutant p53 translation.  The study is well-designed and supported by a coherent set of experiments. However, a few minor points should be addressed to strengthen the manuscript.

Comments

  1. Quantification data are needed for the colony formation assays in Figure 1.
  2. Western blot data throughout the manuscript should be quantified.
  3. Statistical analysis is missing in Figure 2H and 2J.
  4. Clinical relevance of the findings should be discussed. Given the potential translational impact, the authors should elaborate on how these findings might inform future therapeutic strategies or patient stratification.
  5. The study shows an association between CDK4, RBM38, and mutant p53 in TNBC in supplemental Figure 2. The authors should discuss whether similar associations are observed, or could be expected, in other types of cancer where mutant p53 and CDK4 are altered.

Author Response

Response to the reviewer #1

In the manuscript entitled “CDK4/6 inhibitors suppress RB-null triple negative breast cancer by inhibiting mutant p53 expression via RBM38 RNA-binding protein”, Zhang et al has found that CDK4 inhibitors suppress mutant p53 expression in TNBC, especially in Rb-deficient TNBC.  Mechanistically, the authors found that CDK4 phosphorylates RNA-binding protein RBM38, which subsequently modulates mutant p53 translation.  The study is well-designed and supported by a coherent set of experiments. However, a few minor points should be addressed to strengthen the manuscript.

Comments

1. Quantification data are needed for the colony formation assay in Figure 1.

Response: We thank the reviewer’s comment.  The colony formation assay has been quantified in the revised manuscript.

2. Western blot data throughout the manuscript should be quantified.

Response: We thank the reviewer’s comment. We have included densitometry readings/intensity ratio of each band for all the western blots along with uncropped blot in the supplementary data.

3. Statistical analysis is missing in Figure 2H and 2J.

Response: We thank the reviewer’s comment.  Statistical analysis has been added in Figure 2H and 2J in the revised manuscript.

4. Clinical relevance of the findings should be discussed. Given the potential translational impact, the authors should elaborate on how these findings might inform future therapeutic strategies or patient stratification.

Response: We thank the reviewer’s comment.   We have discussed the potential clinical application of the findings.

5. The study shows an association between CDK4, RBM38, and mutant p53 in TNBC in supplemental Figure 2. The authors should discuss whether similar associations are observed, or could be expected, in other types of cancer where mutant p53 and CDK4 are altered.

Response: We thank the reviewer’s comment.   In addition to breast cancer, we also found similar association among CDK4, RBM38, and mutant p53 in liver cancer.

Reviewer 2 Report

Comments and Suggestions for Authors

Summary

This study investigates the role of mutant P53 in triple-negative breast cancer (TNBC) in response to CDK4/6 inhibitor treatment. Using multiple human TNBC cell lines (both RB-proficient and deficient), the study demonstrates that CDK4 phosphorylates RBM38 at serine 195, promoting translation of mutant P53 mRNA. CDK4/6 inhibitors suppress mutant P53 mRNA translation by blocking this phosphorylation, revealing a clear mechanistic link. Overall, the experimental design is robust with a well-established mechanism. There are some minor revisions recommended:

Major points:

  1. Original blots are all cropped. Can you provide the uncropped original blots with ladders? Controls like GAPDH and the protein of interest are highly recommended to be shown on the same blot.
  2. 2: Blotting for total RB in MDA-MB-231 cells is strongly recommended to assess whether RB levels are affected by varying drug dosages and to verify that CDK4/6 inhibitor effects on pRB are not due to altered RB protein levels.

Minor points:

  1. Page 1, last sentence (“As a result … immune checkpoint inhibitors, and targeted agents to enhance the efficacy of CDK4/6 inhibitors in breast and non-breast cancers (refs).”): References are needed to support the claim.
  2. 1 legend: The drug concentrations are reported as palbociclib (2 and 10 mM) and ribociclib (0.5 and 5 mM). Please verify the drug units to match the figure.
  3. 2: Have you examined the level of P21 expression, given its role in downstream target of P53, the level of which could validate the functional impact of mutant P53 suppression?
  4. 2: Font style and size should be consistent across panels. For example, in panel C, “p53” appears larger than the other labels. Additionally, protein names should be presented in all capital letters (e.g., P53 instead of p53). pRb and pRB are all present. Similar applies to Figure.4, Rbm38 and RBM38 need to be consistent.
  5. Page 4, Results section: The sentence “It is well-established that mutant p53-carrying tumor cells, including TNBC cells, are addicted to mutant p53” is not very clear. Please rephrase for precision.
  6. Page 4, last paragraph, font size needs to be consistent.
  7. For the colony formation assay and immunoblot, please provide the number of replicates performed.
  8. Figure 2 H and 2J, the statistical values are missing.

Comments on the Quality of English Language

The manuscript is well-written, with a clear and logical flow of ideas.

Author Response

Response to Reviewer#2

This study investigates the role of mutant P53 in triple-negative breast cancer (TNBC) in response to CDK4/6 inhibitor treatment. Using multiple human TNBC cell lines (both RB-proficient and deficient), the study demonstrates that CDK4 phosphorylates RBM38 at serine 195, promoting translation of mutant P53 mRNA. CDK4/6 inhibitors suppress mutant P53 mRNA translation by blocking this phosphorylation, revealing a clear mechanistic link. Overall, the experimental design is robust with a well-established mechanism. There are some minor revisions recommended:

Major points:

1. Original blots are all cropped. Can you provide the uncropped original blots with ladders? Controls like GAPDH and the protein of interest are highly recommended to be shown on the same blot.

Response: We thank the reviewer’s comment.  The original uncropped blots have been provided and labeled with ladders in the supplemental materials.

2. Blotting for total RB in MDA-MB-231 cells is strongly recommended to assess whether RB levels are affected by varying drug dosages and to verify that CDK4/6 inhibitor effects on pRB are not due to altered RB protein levels.

Response: We thank the reviewer’s comment.  The total RB was measured in MDA-MB-231 cells (Supplemental Figure 1A).

Minor points:

1. Page 1, last sentence (“As a result … immune checkpoint inhibitors, and targeted agents to enhance the efficacy of CDK4/6 inhibitors in breast and non-breast cancers (refs).”): References are needed to support the claim.

Response: We thank the reviewer’s comment.  The references have been added.

2. legend: The drug concentrations are reported as palbociclib (2 and 10 mM) and ribociclib (0.5 and 5 mM). Please verify the drug units to match the figure.

Response: We thank the reviewer’s comment. The typos in the figure legend have been corrected.

3. Have you examined the level of P21 expression, given its role in downstream target of P53, the level of which could validate the functional impact of mutant P53 suppression?

Response: We thank the reviewer’s comment.  We did not test p21 expression in this study. Nevertheless, CDK4/6 inhibitors are known to regulate p21 expression in both p53-dependent and -independent manners (PMID: 33342707; PMID: 34985783)

4. Font style and size should be consistent across panels. For example, in panel C, “p53” appears larger than the other labels. Additionally, protein names should be presented in all capital letters (e.g., P53 instead of p53). pRb and pRB are all present. Similar applies to Figure.4, Rbm38 and RBM38 need to be consistent.

Response: We thank the reviewer’s comment.  The font style has been corrected and is now consistent throughout the manuscript.

5. Page 4, Results section: The sentence “It is well-established that mutant p53-carrying tumor cells, including TNBC cells, are addicted to mutant p53” is not very clear. Please rephrase for precision.

Response: We thank the reviewer’s comment. We have modified the sentence indicating that tumor cells are addicted to mutant p53 for their survival.

6. Page 4, last paragraph, font size needs to be consistent.

Response: We thank the reviewer’s comment. The font size has been corrected.

7. For the colony formation assay and immunoblot, please provide the number of replicates performed.

Response: We thank the reviewer’s comment. Colony formation assays were performed in triplicates.  The western blot analyses are repeated at least two times.

8. Figure 2 H and 2J, the statistical values are missing.

Response: We thank the reviewer’s comment. The statistical analyses have been added in Figure 2H and 2J.